# Prescribed Performance Non-Singular Fast Terminal Sliding Mode Control Based on Extended State Observer for a Deep-Sea Electric Oil-Filled Joint Actuator

Lihui Liao [1] , Baoren Li [1], Yuanyuan Wang [2], Tengfei Tang [3], Dijia Zhang [1] and Gang Yang [1,*]

1    School of Mechanical Science & Engineering, Huazhong University of Science and Technology, Wuhan 430074, China; liaolihuiysu@163.com (L.L.); lbr@hust.edu.cn (B.L.); zhangdijia323@163.com (D.Z.)
2    FESTO Pneumatics Centre, Huazhong University of Science and Technology, Wuhan 430074, China; wyuan2011@163.com
3    College of Mechanical and Electrical Engineering, Wuhan Institute of Technology, Wuhan 430074, China; t.tfei@hotmail.com
*    Correspondence: ygxing73@hust.edu.cn

**Abstract:** High dynamic performance of a deep-sea electric oil-filled joint actuator is an important premise to guarantee the working performance of an electric underwater manipulator. However, the unfavorable factors (i.e., extremely high water pressure, near freezing temperature) brought by the deep-sea working environment seriously affect the characteristic and dynamic performance of the electric oil-filled joint actuator, which mainly includes oil stirring viscos loss, output shaft dynamic seal loss, and core loss. In this paper, a novel observer-based robust control method named prescribed performance non-singular fast-terminal sliding-mode control (PP-NFTSMC-ESO) was synthesized for improving the dynamic performance of a deep-sea electric oil-filled joint actuator. The extended state observer (ESO) was employed to observe the unmeasured joint velocity signal and estimate the lumped uncertainties, while the prescribed performance function (PPF) was applied to constrain the instantaneous and steady-state performance of the trajectory-tracking error. The robust NFTSMC control method was then established by integrating the function of ESO and PPF through backstepping methodology. The stability of the proposed PP-NFTSMC-ESO strategy was analyzed and proved by the Lyapunov's stability theory. It was proven that under the proposed controller, all the closed-loop signals are bounded and the trajectory tracking errors will converge to a small neighborhood of the origin with appropriate design parameters. The effectiveness of the proposed control scheme was illustrated by comparative simulation studies.

**Keywords:** deep-sea electric oil-filled joint actuator; robust control; non-singular fast-terminal sliding mode control (NFTSMC); extended state observer (ESO); prescribed performance control

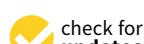



## 1. Introduction

Recently, with the exhaustion of land resources, underwater resources are at the center of attention, especially the resources hidden in the deep sea [1]. Therefore, many underwater works and operations are now being performed both in the scientific and business communities, such as salvage, marine resource investigation, ship engineering, marine construction, etc. [2]. Owing to the extremely poor working conditions in the deep sea, underwater manipulators, often equipped on the remotely operated underwater vehicles or autonomous and remotely operated underwater vehicles, are considered to be the most suitable tool to work instead of human beings. Hydraulic underwater manipulators and electric underwater manipulators are the most commonly existing commercially available underwater manipulators and most of the experimental/prototype underwater manipulators [3]. In recent years, with the development and combination of artificial intelligence technology and robot technology, the research of electric underwater manipulators has

attracted considerable attention due to their capability for precise motion and force/torque control as they perform in the industrial fields [3]. Typical electric underwater manipulators are the manipulator 7E of Eca Robotics [3], the UMA manipulator developed by Graal Tech SRL in Italy for the TRIDENT project [4], the modified commercial electric manipulator ARM 5E [5], etc.

Owing to the extremely high pressure in the deep-sea environment, the electric underwater manipulator is watertight and oil compensated from bearing the deep-sea water pressure [6,7]. The oil-filled joint actuator of electric underwater manipulator, which is commonly composed of a brushless, direct-drive motor with reduction gearbox featuring low backlash and a large reduction ratio [3], will suffer from oil stirring loss [7–10] resulting from the rotation viscos between the high speed rotor and oil, output shaft dynamic seal loss [10] deriving from the high-pressure action and high-speed rotation friction on the rubber seal rings, and core loss [9,10] caused by the high-pressure action on the motor cores. Therefore, the oil-filled joint actuator will show different characteristics and response performance compared with its common use.

For the oil-filled joint actuator, the oil stirring viscos loss and output shaft dynamic seal loss can be considered as part of unknown internal disturbance and external disturbance respectively, while the core loss usually directly leads to the physical parameter deviations of the motor, which can be treated as dynamic uncertainties. Furthermore, due to the compact and lightweight requirements, usually only the angular position sensor is available for the oil-filled joint actuator, and thus joint velocity is immeasurable for the control. Consequently, the oil-filled joint actuator will suffer from an unmeasured system state, dynamic uncertainties, and unknown disturbances when it works in the deep-sea environment, and its achievable control performance could be severely deteriorated by these adverse factors.

To handle immeasurable system states, observers have been widely used [11–15]. In [11], a neural-based full-order Luenberger adaptive observer was designed for sensorless linear induction motor control. In [12], a sliding mode observer was proposed to observe the back electromotive force for obtaining the velocity and position of the mover of a permanent magnet synchronous linear motor. In [13], an approximate high-gain observer was employed to observe the speed signals for an induction motor control. In [14], a third-order nonlinear extended-state observer (ESO) was constructed for position and speed estimation for a permanent magnet synchronous motor control. In [15], the oxygen excess ratio was estimated via an extended-state observer (ESO) from the measurements of the compressor flow rate, the load current, and the supply manifold pressure, which was used in the output feedback controller design of the oxygen excess ratio control system.

To reduce the effect of dynamic uncertainties, adaptive-based controllers are the most commonly used methods [16–24]. For example, in [17], an adaptive robust controller with ESO (ARCESO) was synthesized for high-accuracy motion control of a DC motor, in which the adaptive control was presented to deal with for the parametric uncertainty. To suppress the parametric uncertainty, a neural network learning adaptive robust controller was synthesized for an industrial linear motor stage to achieve good tracking performance and excellent disturbance rejection ability, where the parametric variations were handled by the adaptation part of the controller [18]. Although adaptive based controllers can achieve satisfactory performance in many physical systems, the parameter variation range should be known in advance [17,18,23], which is usually difficult to acquire in advance of practical application, thus leading to the limitation of wide application.

To overcome the effect of unknown disturbances, disturbance observers are intuitive and effective methods [17,18,25–30]. In [16], an ESO was constructed to estimate the unstructured uncertainties (including nonlinear friction, external disturbances, and unmodeled dynamics) for designing an ARCESO high-accuracy motion controller of a DC motor. In [25], a generalized proportional integral observer was designed for dealing with load torque disturbance and time-varying parameter uncertainties for the finite control set predictive current control of induction motor systems. Due to the universal approximation

feature, neural networks (NNs) and fuzzy logic systems (FLSs) are also widely used to deal with the unknown disturbances. To approximate the unknown disturbances, a NNs learning algorithm was employed to design the adaptive robust controller for an industrial linear motor stage to achieve good tracking performance and excellent disturbance rejection ability [18]. FLSs were utilized to approximate the unknown disturbances of a bionic mechanical leg for the adaptive fuzzy robust controller design [28] in the previous work of the authors in this paper. In addition, robust control has also been a choice to some researchers to attenuate disturbances in physical systems, such as active disturbance rejection control [31,32], sliding-mode control (SMC)-based methods [33–44], etc. Among them, the nonsingular fast terminal sliding-mode control (NFTSMC) [30,38–44], as a new typical robust controller, has been widely used in controlling uncertain systems because of its attractive properties such as fast dynamic response, robustness against uncertainties, chattering phenomenon elimination, finite time convergence, and its simple design procedure [30,38].

Motivated by the above observations, to provide a high-performance motion controller with capabilities of unmeasured system states self-estimating, dynamic uncertainties and unknown disturbances rejection for a deep-sea electric oil-filled joint actuator, a novel extended-state observer-based prescribed performance non-singular fast-terminal sliding-mode control (PP-NFTSMC-ESO) strategy was proposed in this paper. The ESO was utilized to estimate the unmeasured system states and the lumped uncertainties to make the precise model-based compensation, while the residual parts including estimation errors and stable tracking errors were discharged by the simple robust term and stable feedback term, respectively. In order to improve the transient and steady-state position responses, an error constraint transformation was developed to guarantee the prescribed time-varying performance. The main contributions of this paper can be summarized as follows:

(1) A novel PP-NFTSMC-ESO controller was proposed for high-performance motion control of a deep-sea electric oil-filled joint actuator in the presence of unmeasured system state, dynamic uncertainties, and unknown disturbances, which combines the advantages of NFTSMC control in terms of robustness against uncertainties, chattering phenomenon elimination, fast dynamic response, finite time convergence, and simple design procedure, the restraining action of prescribed performance control for the transient and steady state performance, and the excellent observation ability of the ESO for system states and lumped uncertainties.

(2) With the proposed control method, the mechanical configuration of the deep-sea electric oil-filled joint actuator can be simplified with only an angular position sensor, which benefits for the structure design and electrical design, but with no performance deterioration of trajectory tracking control.

(3) The stability of the proposed controller was theoretically proven by the Lyapunov stability theory. The excellent trajectory tracking performance was demonstrated with the studies on different working conditions, and the superiority of the proposed controller was illustrated by the comparison with proportional-integral (PI) controller, ESO-based sliding-mode controller (SMC-ESO), and ESO-based non-singular fast-terminal sliding-mode controller (NFTSMC-ESO).

The remainder of this paper is organized as follows. Section 2 presents the dynamic modeling and problem formulation. Section 3 introduces the ESO. The design procedure of the PP-NFTSMC-ESO control method and the stability proving process of this controller-observer strategy are described in Section 4. The effectiveness is demonstrated via simulation studies in Section 5, and conclusions are provided in Section 6.

## 2. Dynamic Modeling and Problem Formulation

### 2.1. Dynamic Modeling

Figure 1 shows the structure of a deep-sea electric oil-filled joint actuator, which is a test joint of the mechanic leg of a deep-sea crawling robot named "Qilin". The oil-filled

joint actuator is mainly composed of a motor, a gear reducer, an angular position sensor, a housing with sealing assembly, and a pressure compensation system.

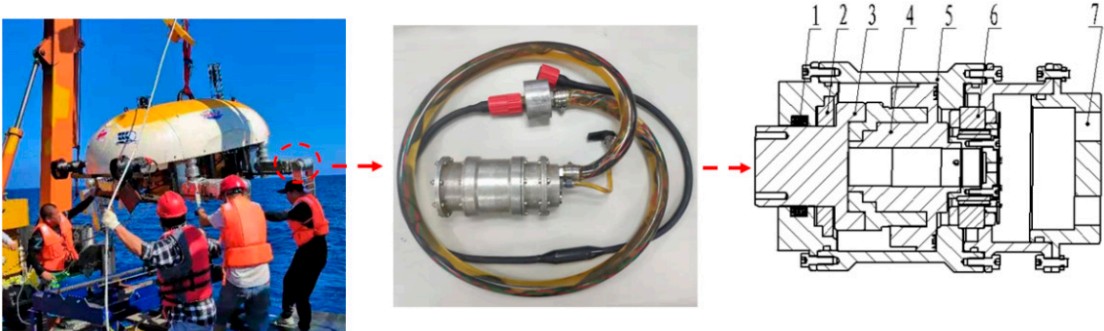

**Figure 1.** Structure of a deep-sea electric oil-filled joint actuator. 1—sealing assembly; 2—angular position sensor; 3—output shaft; 4—gear reducer; 5—housing; 6—motor; 7—pressure compensation interface.

The dynamic model of the deep-sea electric oil-filled joint actuator can be described as follows:

$$J(N\ddot{q}) = k_t u - k_b(N\dot{q}) - T_v - \frac{T_d}{N} - \frac{T_f}{N} \tag{1}$$

where $q$, $\dot{q}$, $\ddot{q}$ are joint actuator output position, velocity, and acceleration, respectively; $u$ is the control current of the motor; $J$, $k_t$, $k_b$, and $N$ are the moment of inertia, the motor torque constant, the motor viscous friction coefficient, and the gear reduction ratio, respectively; $T_v$, $T_d$, and $T_f$ are oil stirring loss torque, unknown external disturbance torque, and dynamic seal loss torque of the joint actuator output shaft, respectively.

Define $\theta_1 = k_t/(JN)$, $\theta_2 = -k_b/J$, then the dynamic model (1) of the joint actuator can be expressed as follows:

$$\ddot{q} = \theta_1 u + \theta_2 \dot{q} + \frac{1}{J}\left(-\frac{T_v}{N} - \frac{T_d}{N^2} - \frac{T_f}{N^2}\right) \tag{2}$$

As the precise values of $J$, $k_t$, and $k_b$ are usually unknown in practical applications, the parameters $\theta_1$ and $\theta_2$ can be expressed as the combination of nominal values and uncertain values as follows:

$$\begin{cases} \theta_1 = \theta_{1n} + \Delta\theta_1 \\ \theta_2 = \theta_{2n} + \Delta\theta_2 \end{cases} \tag{3}$$

where $\theta_{1n} = k_{tn}/(J_n N)$ and $\theta_{2n} = -k_{bn}/J_n$ are determined by the nominal values of the joint actuator (denoted by $J_n$, $k_{tn}$, and $k_{bn}$), while $\Delta\theta_1$ and $\Delta\theta_2$ represent deviations from the nominal parameters of the system model.

Substituting Equation (3) into (2), the system model can be expressed as follows:

$$\ddot{q} = \theta_{1n} u + \theta_{2n} \dot{q} + \left[\Delta\theta_1 u + \Delta\theta_2 \dot{q} + \frac{1}{J}\left(-\frac{T_v}{N} - \frac{T_d}{N^2} - \frac{T_f}{N^2}\right)\right] \tag{4}$$

Defining $x = [x_1, \ x_2]^T = [q, \ \dot{q}]^T$ as the system state vector, the dynamic Equation (4) can be rewritten in a state-space form:

$$\begin{cases} \dot{x}_1 = x_2 \\ \dot{x}_2 = \theta_{1n} u + \theta_{2n} x_2 + d \\ y = x_1 \end{cases} \tag{5}$$

where $y$ is the output of the joint actuator, and $d = \Delta\theta_1 u + \Delta\theta_2 \dot{q} + \frac{1}{J}\left(-\frac{T_v}{N} - \frac{T_d}{N^2} - \frac{T_f}{N^2}\right)$ represents the lumped uncertainties of the joint actuator, including parametric uncertainties and unknown internal and external disturbances.

### 2.2. Oil Stirring Viscos Loss Modeling

Due to high pressure in the deep sea, deep-sea electric oil-filled joint actuator will suffer from oil stirring viscos loss, output shaft dynamic seal loss, and core loss. Since the cause and change rules of output shaft dynamic seal loss and core loss are very complicated, there are no strict expressions at present describing these two losses, and we only focus on the modeling of oil stirring viscos loss.

Many researchers have studied the cause and change rules of the oil stirring viscos loss. The impact of oil type and motor structure on the oil stirring viscos loss was studied in [7]. The influence of temperature and pressure on the change rules of the oil stirring viscos loss was observed in [8]. In [9], the impact of pressure on the oil stirring viscos loss was researched. As the motor structure and oil type are not changed after the oil-filled joint actuator molding, we adopted the change rules summarized by reference [8] to calculate the oil stirring viscos loss in this paper.

According to reference [8], the oil stirring viscos loss of an oil-filled motor is a combination of the flank viscos loss and the disk viscos loss, which is expressed as

$$T = T_{v1} + T_{v2} = \frac{\pi L d^3 \mu \omega}{4\delta} + \frac{\pi\left(d_2^4 - d_1^4\right)\mu\omega}{32h} = k\mu\omega \tag{6}$$

$$\mu = \mu_0 \exp(\alpha P) \tag{7}$$

where $k = \frac{\pi L d^3}{4\delta} + \frac{\pi\left(d_2^4 - d_1^4\right)}{32h}$ is a constant related to the motor geometry; $d$ is the diameter of the rotor, $d_1$ is the diameter of the rotor end face, $d_2$ is the diameter of the end face of the motor housing, and $h$ is the distance between the end face of the rotor and the end face of the motor housing; $\omega$ is the angular velocity, $P$ is oil pressure, $\mu$ is the dynamic viscosity, $\mu_0$ is the dynamic viscosity at given temperature $T_0$ and pressure $P_0$, and $\alpha$ is the viscos-pressure coefficient.

In general, the viscosity of oil is related to pressure and temperature, increasing with pressure and decreasing with temperature, which is a complicated changing rule. Therefore, researchers usually obtained the corresponding values of $\mu_0$ and $\alpha$ by experiments corresponding to the typical working conditions.

### 2.3. Prescribed Performance Function

To constrain the transient and steady-state performances of the system error $e(t)$, define $e(t)$ satisfies strictly the following inequality [45,46]:

$$-l\rho(t) < e(t) < h\rho(t) \tag{8}$$

where $\rho(t) = (\rho_0 - \rho_\infty)e^{-mt} + \rho_\infty$ is the selected positive decreasing and bounded prescribed performance function, $\rho_0$, $\rho_\infty$ and $m$ are positive design parameters, and $\rho_\infty = \lim_{t \to \infty} \rho(t) < \rho_0$, $l$ and $h$ are positive parameters to be designed.

To simplify the designing of prescribed performance controller, a strictly monotonic increasing function $S(z(t))$ is introduced to transform the performance constrained problem into an unconstrained stabilization problem.

$$S(z(t)) = \frac{h\exp(z) - l\exp(-z)}{\exp(z) + \exp(-z)} \tag{9}$$

Consequently, condition (8) can be rewritten as

$$e(t) = \rho(t)S(z(t)) \tag{10}$$

and the transformed error $z(t)$ can be derived as

$$z(t) = S^{-1}\left(\frac{e(t)}{\rho(t)}\right) = \frac{1}{2}\ln\frac{e(t)/\rho(t) + l}{h - e(t)/\rho(t)} \qquad (11)$$

The time derivative of $z(t)$ results in

$$\dot{z}(t) = r\left(\dot{e}(t) - \frac{\dot{\rho}(t)e(t)}{\rho(t)}\right) \qquad (12)$$

where $r = \frac{1}{2\rho(t)}\left(\frac{1}{z(t)+l} - \frac{1}{z(t)-h}\right)$.

**Lemma 1** [47]: *For any constant $\tau > 0$ and variable $B \in R$, there exists equation $\lim\limits_{B \to 0}\frac{\tanh(B/\tau)}{B} = 0$; Let $\theta\{\varepsilon||\varepsilon| > 0.8814\tau\}$, then for $\forall \varepsilon \in \theta$, the following inequality holds*

$$1 - 2\tanh^2\left(\frac{\varepsilon}{\tau}\right) < 0 \qquad (13)$$

## 3. Extended State Observer Design

Designing a high-performance model-based controller usually requires a full-state feedback. However, for the electric oil-filled joint actuator, only the joint position sensor is installed in the joint. Therefore, the task of the ESO observer is to estimate the unmeasured system state $x_2$ and the lumped uncertainty $d$ for the later controller design.

ESO has many advantages than other observers. Besides fast convergence and high robustness characteristics, the most important thing to appreciate is that it can estimate both system states and the lumped uncertainties simultaneously with a boundedness of the estimation errors. Therefore, ESO is becoming an effective tool in the control of dynamic systems.

According to the structure of the system model (5), extend the lumped uncertainty $d$ as state $x_3$, and $f(t)$ represent the time derivative of $d$, then the dynamic Equation (5) can be rewritten as

$$\begin{cases} \dot{x}_1 = x_2 \\ \dot{x}_2 = \theta_{1n}u + \theta_{2n}x_2 + x_3 \\ \dot{x}_3 = f(t) \end{cases} \qquad (14)$$

**Assumption 1**: *$f(t)$ is bounded, i.e., $|f(t)| \leq \overline{f}$, where $\overline{f} > 0$ is a constant.*

According to Equation (14), the ESO is designed as

$$\begin{cases} \dot{\hat{x}}_1 = \hat{x}_2 + l_1\omega_0\tilde{x}_1 \\ \dot{\hat{x}}_2 = \theta_{1n}u + \theta_{2n}\hat{x}_2 + \hat{x}_3 + l_2\omega_0^2\tilde{x}_1 \\ \dot{\hat{x}}_3 = l_3\omega_0^3\tilde{x}_1 \end{cases} \qquad (15)$$

where $\hat{x} = [\hat{x}_1,\ \hat{x}_2,\ \hat{x}_3]^T$ is the estimated state vector, $\tilde{x} = x - \hat{x} = [\tilde{x}_1,\ \tilde{x}_2,\ \tilde{x}_3]^T$ is the estimation error vector, and $\omega_0 > 0$ can be treated as the bandwidth of the ESO.

In order to guarantee the stability of the ESO, the gains of the observer are designed to satisfy the following Hurwitz polynomial [48,49]:

$$L(s) = s^3 + l_1\omega_0 s^2 + l_2\omega_0^2 s + l_3\omega_0^3 = (s + 5\omega_0)^3 \qquad (16)$$

According to Equations (14) and (15), the dynamic equation of the estimation errors is represented as

$$\begin{cases} \dot{\tilde{x}}_1 = \tilde{x}_2 - l_1\omega_0\tilde{x}_1 \\ \dot{\tilde{x}}_2 = \theta_{2n}\tilde{x}_2 + \hat{x}_3 - l_2\omega_0^2\tilde{x}_1 \\ \dot{\hat{x}}_3 = f(t) - l_3\omega_0^3\tilde{x}_1 \end{cases} \qquad (17)$$

**Theorem 1** [17,48]: *Under Assumption 1, the estimated states are always bounded and there exist a constant $\sigma_i > 0$, and a finite time $t_1 > 0$ such that*

$$|\tilde{x}_i| \leq \sigma_i, \ \sigma_i = O\left(\frac{1}{\omega_0^c}\right). \ i = 1, 2, 3, \ \forall t > t_1 \tag{18}$$

*for some positive integer c.*

From Theorem 1, we can see that the estimation errors $\tilde{x}_i$ are bounded and will converge to an arbitrarily small range if the parameter $\omega_0$ increases largely enough, i.e., $\lim_{\omega_0 \to \infty} \hat{x}_i = x_i, \ i = 1, 2, 3.$

### 4. PP-NFTSMC-ESO Controller Design

For the trajectory tracking control of a deep-sea electric oil-filled joint actuator with unmeasured system states, dynamic uncertainties, and unknown disturbances, the PP-NFTSMC-ESO controller was proposed in this paper. The block diagram of this controller is shown in Figure 2. The ESO was applied to estimate the unmeasured velocity and the lumped uncertainty, while the PPF was employed to constrain the transient and steady-state performances of the trajectory tracking error. The NFTSMC controller was then synthesized with the signals from the reference trajectory, ESO, and PPF.

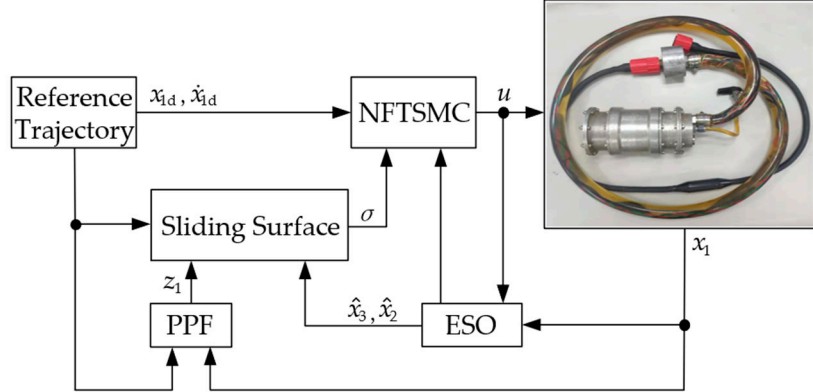

**Figure 2.** The block diagram of the proposed PP-NFTSMC-ESO controller.

Define the position error $e$ and velocity error $z_2$ as

$$e = y - y_d = x_1 - x_{1d} \tag{19}$$

$$z_2 = \hat{x}_2 - \alpha \tag{20}$$

where $y_d$ and $x_{1d}$ are the desired joint position, and $\alpha$ is the virtual control function of $x_2$.

Step 1: According to the results of the prescribed performance function (11), the position error $e$ can be transformed as follows

$$z_1 = \frac{1}{2} \ln \frac{e/\rho + l}{h - e/\rho} \tag{21}$$

The time derivative of $z_1$ is

$$\dot{z}_1 = r\left(\dot{e} - \frac{\dot{\rho}e}{\rho}\right) = r\left(x_2 - \dot{x}_{1d} - \frac{\dot{\rho}e}{\rho}\right) \tag{22}$$

Therefore, the virtual control function of $x_2$ can be designed as

$$\alpha = -k_0 z_1 + \dot{x}_{1d} + \frac{\dot{\rho} e}{\rho} \tag{23}$$

where $k_0$ is a positive design parameter.

From Equation (20), we can have $x_2 = z_2 + \alpha + \widetilde{x}_2$, and substituting it into Equation (22) results in

$$\dot{z}_1 = r\left(z_2 + \alpha + \widetilde{x}_2 - \dot{x}_{1d} - \frac{\dot{\rho} e}{\rho}\right) \tag{24}$$

Define the Lyapunov function as

$$V_1 = \frac{1}{2r} z_1^2 \tag{25}$$

Substituting Equations (23) and (24) into (25), the time derivative of $V_1$ is

$$\dot{V}_1 = \frac{1}{r} z_1 \dot{z}_1 = -k_0 z_1^2 + z_1 z_2 + z_1 \widetilde{x}_2 \tag{26}$$

Step 2: Define NFTSMC sliding function as [30]

$$\sigma = z_2 + \int_0^t \left(k_1 |z_1|^{[\alpha_1]} + k_2 |z_2|^{[\alpha_2]}\right) d\tau \tag{27}$$

where $\alpha_1$, $\alpha_2$ are positive constants subject to $0 < \alpha_1 < 1$ and $\alpha_1 < \alpha_2$, $|z_i|^{[\alpha_i]} = |z_i|^{\alpha_i} \mathrm{sgn}(z_i)$, $i = 1, 2$; $k_1$ and $k_2$ are positive design parameters.

The time derivative of $\sigma$ is

$$\dot{\sigma} = \dot{\hat{x}}_2 - \dot{\alpha} + k_1 |z_1|^{[\alpha_1]} + k_2 |z_2|^{[\alpha_2]} = \dot{x}_2 - \dot{\widetilde{x}}_2 - \dot{\alpha} + k_1 |z_1|^{[\alpha_1]} + k_2 |z_2|^{[\alpha_2]} \tag{28}$$

Substituting the second equation of (13) and (14) into (29) produces

$$\dot{\sigma} = \theta_{1n} u + \theta_{2n} x_2 + x_3 - \dot{\alpha} - \theta_{2n} \widetilde{x}_2 - \widetilde{x}_3 + l_2 \omega_0^2 \widetilde{x}_2 + k_1 |z_1|^{[\alpha_1]} + k_2 |z_2|^{[\alpha_2]} \tag{29}$$

Define the Lyapunov function as

$$V_2 = V_1 + \frac{1}{2}\sigma^2 \tag{30}$$

Substituting Equations (26) and (29) into (30), the time derivative of $V_2$ is

$$\begin{aligned} \dot{V}_2 = \quad & \dot{V}_1 + \sigma\dot{\sigma} = -k_0 z_1^2 + z_1 z_2 + z_1 \widetilde{x}_2 \\ & + \sigma\left[\theta_{1n} u + \theta_{2n} x_2 + x_3 - \dot{\alpha} - \theta_{2n} \widetilde{x}_2 - \widetilde{x}_3 + l_2 \omega_0^2 \widetilde{x}_1 + k_1 |z_1|^{[\alpha_1]} + k_2 |z_2|^{[\alpha_2]}\right] \end{aligned} \tag{31}$$

The control $u$ of PP-NFTSMC-ESO controller is designed as

$$u = -\frac{1}{\theta_{1n}}\left(u_{eq} + u_{sw} + u_b\right) \tag{32}$$

$$u_{eq} = \theta_{2n} \hat{x}_2 + \hat{x}_3 - \dot{\alpha} + k_1 |z_1|^{[\alpha_1]} + k_2 |z_2|^{[\alpha_2]} \tag{33}$$

$$u_{sw} = \eta\,\mathrm{sign}(\sigma) + k_3 \sigma \tag{34}$$

$$u_b = 2\frac{|z_1 z_2|}{\sigma}\tanh^2\left(\frac{\sigma}{\tau}\right) \tag{35}$$

where $\eta$, $\tau$, and $k_3$ are positive design parameters.

As seen, the PP-NFTSMC-ESO controller consists of three parts, where $u_{eq}$ is the equivalent control law to hold the error trajectory on the sliding surface, $u_{sw}$ is the robust

term used to compensate for the estimation errors, and $u_b$ is the stable feedback term to stable the tracking error.

Substituting Equations (32)–(35) into (31) results in

$$\dot{V}_2 = \dot{V}_1 + \sigma\dot{\sigma} = -k_0 z_1^2 + z_1\tilde{x}_2 + l_2\omega_0^2\tilde{x}_1\sigma + z_1 z_2 - 2|z_1 z_2|\tanh^2\left(\frac{\sigma}{\tau}\right) - k_3\sigma^2 - \eta|\sigma| \quad (36)$$

According to Lemma 1, it follows that

$$\dot{V}_2 \leq -k_0 z_1^2 + z_1\tilde{x}_2 + l_2\omega_0^2\tilde{x}_1\sigma - k_3\sigma^2 - \eta|\sigma| \quad (37)$$

Applying Cauchy-Schwarz inequality, (37) can be transformed to

$$
\begin{aligned}
\dot{V}_2 &\leq -k_0 z_1^2 + \tfrac{1}{2}\omega_0^2 z_1^2 + \tfrac{1}{2\omega_0^2}\tilde{x}_2^2 + \tfrac{1}{2}\left(l_2\omega_0^2\tilde{x}_1\right)^2 + \tfrac{1}{2}\sigma^2 - k_3\sigma^2 + \tfrac{1}{2}\sigma^2 + \tfrac{1}{2}\eta^2 \\
&\leq -k_0 z_1^2 + \tfrac{1}{2}\omega_0^2 z_1^2 - k_3\sigma^2 + \sigma^2 + \tfrac{1}{2\omega_0^2}\tilde{x}_2^2 + \tfrac{1}{2}\left(l_2\omega_0^2\tilde{x}_1\right)^2 + \tfrac{1}{2}\eta^2 \\
&= -\left(k_0 - \tfrac{1}{2}\omega_0^2\right)z_1^2 - (k_3 - 1)\sigma^2 + \Omega \\
&\leq -\gamma V_2 + \Omega
\end{aligned}
\quad (38)
$$

where $\gamma = \min\left\{2r\left(k_0 - \tfrac{1}{2}\omega_0^2\right), \; 2(k_3 - 1)\right\}$, and $\Omega = \tfrac{1}{2\omega_0^2}\tilde{x}_2^2 + \tfrac{1}{2}l_2^2\omega_0^4\tilde{x}_1^2 + \tfrac{1}{2}\eta^2$.

As $\Omega$ is a function of the estimation errors, which will converge to a small value after a finite time $t_1$ when selected a large enough $\omega_0$. Therefore, if letting $k_0 > \tfrac{1}{2}\omega_0^2$, $k_3 > 1$, and $T = t - t_1$, we have

$$V_2 \leq V_2(t_1)\exp(-\gamma T) + \frac{\Omega}{\gamma}[1 - \exp(-\gamma T)], \; \forall t > t_1 \quad (39)$$

**Theorem 2**: *For the deep-sea electric oil-filled joint actuator (5), under Assumption 1, with the ESO (15), the proposed control law (32)–(35) guarantees that the closed-loop system is stable and the output position tracking error converges to a small neighborhood of the origin by appropriately choosing the observer parameter $l_1$, $l_2$, $l_3$, $\omega_0$, and the controller parameters $k_0$, $k_1$, $k_2$, $k_3$, and $\eta$.*

## 5. Numeric Simulation

### 5.1. Simulation Setup

In order to verify the performance of the proposed PP-NFTSMC-ESO controller, computer simulations were conducted based on a simulation model of the electric oil-filled joint actuator, which is shown in Figure 3. The nominal parameters of the simulation model were set with the actual parameters of the electric oil-filled joint actuator listed in Table 1. The simulations were performed with a fixed sampling time 1 ms, and the initial states of the simulation model were set as $x = [0,\ 0]^T$, while that of the ESO was set as $\hat{x} = [0,\ 0,\ 0]^T$.

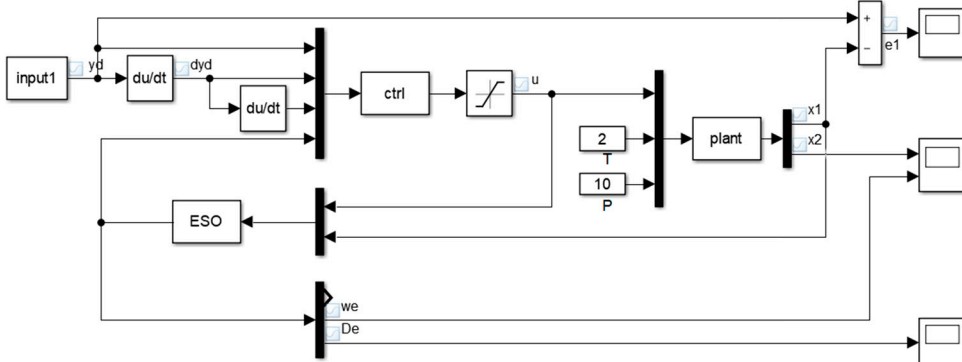

**Figure 3.** Simulation model of the electric oil-filled joint actuator.

**Table 1.** The nominal parameters of the electric oil-filled joint actuator.

| Parameter | $J_n$ (kg m$^2$) | $k_{tn}$ (Nm/A) | $k_{bn}$ (Nm/(rad/s)) | $N$ |
|-----------|------------------|-----------------|----------------------|-----|
| value | $3.34 \times 10^{-5}$ | 0.105 | $1.95 \times 10^{-5}$ | 100 |

The oil stirring viscos loss was set as Equation (6) with the working pressure and temperature in two cases, $P$ = 10 MPa, $T$ = 2 °C, and $P$ = 0 MPa, $T$ = 25 °C, which were used to simulate the deep-sea working condition with 1000 m depth (the temperature of deep sea is generally about 2 °C) and the normal laboratory working condition. The values of $\mu_0$ and $\alpha$ are listed in Table 2 (the oil used in this work was the ISO VG 22 hydraulic oil), and the geometry parameters of the researched electric oil-filled joint actuator were presented in Table 3.

**Table 2.** The values of $\mu_0$ and $\alpha$ corresponding to the working conditions of 25 °C and 2 °C [8].

| Parameter | | $\mu_0$ (mPas) | $\alpha$ [1] |
|-----------|--------|----------------|----------|
| value | 2 °C | 140.3 | $\frac{1}{4.49 \times 10^7 + 1.60 \times 10^{-2}\,P}$ |
| | 25 °C | 41.6 | $\frac{1}{2.01 \times 10^7 + 5.10 \times 10^{-1}\,P}$ |

[1] The unit of $P$ is MPa.

**Table 3.** The geometry parameters of the researched electric oil-filled joint actuator.

| Parameter | $d$ | $d_1$ | $d_2$ | $\delta$ | $h$ | $L$ |
|-----------|-----|-------|-------|----------|-----|-----|
| value | 37.5 mm | 63 mm | 37.5 mm | 0.3 mm | 38 mm | 15.5 mm |

The external disturbance torque, output shaft dynamic seal friction, and the core loss are all unknown but deteriorate the performance of the electric oil-filled joint actuator. In this work, in order to simulate their influence, we set the external disturbance torque as time varying sinusoidal signals, and the influence of core loss as parameter variance of the electric oil-filled joint actuator, which are presented as Equations (40) and (41) respectively. Since the output shaft dynamic seal friction is related to joint speed and working pressure, we set the variation regulation of it as Equation (42), of which the amplitude surface is shown in Figure 4 with the maximum and minimum values are 10 Nm (at the deep-sea working condition $P$ = 10 MPa and $T$ = 2 °C) and 3 Nm (at the normal laboratory working condition $P$ = 0 MPa and $T$ = 25 °C), respectively. Although these regulations are not from their real law of changes, they are able to simulate the influence of the dynamic uncertainties and unknown disturbances on the electric oil-filled joint actuator to some extent.

$$T_d = 20(1 + 0.5\sin(0.5\pi t)) \tag{40}$$

$$\begin{cases} J = J_n(1 + 0.2\sin(0.5\pi t)) \\ k_t = k_{tn}(1 + 0.3\cos(0.3\pi t)) \\ k_b = k_{bn}(1 + 0.2\cos(0.3\pi t)) \end{cases} \tag{41}$$

$$T_f = (1 + \exp(0.3591|x_2|) + \exp(0.1504P))\text{sign}(x_2) \tag{42}$$

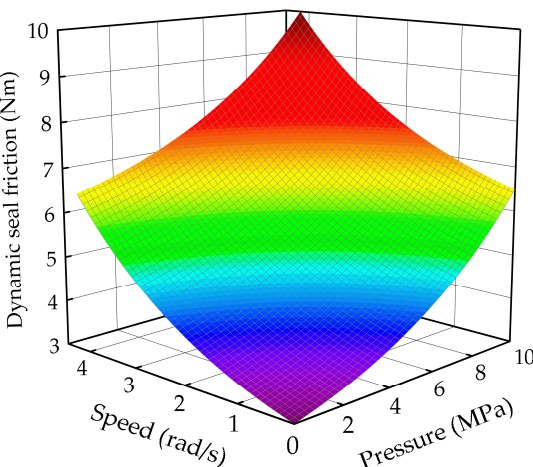

**Figure 4.** Output shaft dynamic seal friction amplitude surface.

*5.2. Controllers for Comparison*

　　To demonstrate the superiority of the proposed PP-NFTSMC-ESO controller, three other controllers were employed for comparison, which are the PI controller, SMC-ESO controller, and NFTSMC-ESO controller.

　　(1) The PI controller is described as

$$u = k_p(x_{1d} - x_1) + k_i \int_0^t (x_{1d} - x_1)d\tau \tag{43}$$

　　(2) The SMC-ESO controller is derived as

$$u = -\frac{1}{\theta_{1n}}\left(u_{eq} + u_{sw}\right) \tag{44}$$

$$u_{eq} = \theta_{2n}\hat{x}_2 + \hat{x}_3 + k_1\left(\hat{x}_2 - \dot{x}_{1d}\right) - \ddot{x}_{1d} \tag{45}$$

$$u_{sw} = \eta\,\text{sign}(\sigma) + k_2\sigma \tag{46}$$

The design procedure is included in Appendix A.

　　(3) The NFTSMC-ESO controller is achieved as

$$u = -\frac{1}{\theta_{1n}}\left(u_{eq} + u_{sw}\right) \tag{47}$$

$$u_{eq} = \theta_{2n}\hat{x}_2 + \hat{x}_3 - \ddot{x}_{1d} + k_1|z_1|^{[\alpha_1]} + k_2|z_2|^{[\alpha_2]} \tag{48}$$

$$u_{sw} = \eta\,\text{sign}(\sigma) + k_2\sigma \tag{49}$$

The design procedure is included in Appendix B.

**Remark 1**: *The parameters of the PI controller are chosen by the PID Turner Toolbox of MATLAB with some fine-tuning due to system parameter fluctuation and disturbances, in which the robustness index is improved as much as possible while ensuring the tracking accuracy. The parameters of SMC-ESO and NFTSMC-ESO controllers were selected by trial and error, in which we chose sufficiently large parameters on the premise of ensuring stability prerequisites. In this way, the response performance of the system will be better guaranteed, and the prerequisites will also be satisfied locally around the desired trajectory to be tracked. This is a practical method and has been used by other researchers to synthesize control methods for dynamic systems [17,23]. In order to make the comparison study fair, the parameters of the proposed PP-NFTSMC–ESO controller were inherited from the NFTSMC-ESO controller, while the additional parameters were obtained by trial and error. The parameter $\omega_0$ of the ESO was chosen according to the method in [48] to ensure the*

*stability of the observer and desired state convergence performance. All the design parameters of the controllers and observers are shown in Table 4.*

**Table 4.** The design parameters of the controllers and observers.

| Controller/Observer | Parameter |
|---|---|
| PI | $k_p = 500,\ k_i = 300$ |
| SMC-ESO | $k_1 = 20,000,\ k_2 = 10,000,\ \eta = 100$ |
| NFTSMC-ESO | $k_1 = 1000,\ k_2 = 20,\ k_3 = 200,\ \alpha_1 = 1.1,\ \alpha_2 = 1.3,\ \eta = 0.1$ |
| PP-NFTSMC-ESO | $k_0 = 100,\ k_1 = 1000,\ k_2 = 20,\ k_3 = 200,\ \alpha_1 = 1.1,\ \alpha_2 = 1.3,$ $\eta = 0.1,\ \tau = 0.06,\ \rho_0 = 0.25,\ \rho_\infty = 0.05,\ m = 0.5,\ l = 1,\ h = 1$ |
| ESO | $l_1 = 15,\ l_2 = 75,\ l_3 = 125,\ \omega_0 = 60$ |

*5.3. Simulation Study*

Considering the actual working conditions, computer simulation studies were performed on the electric oil-filled joint actuator for trajectory tracking control in the presence of unmeasured system states, dynamic uncertainties, and unknown external disturbances.

Case 1: This case is to verify the effectiveness of the proposed PP-NFTSMC-ESO controller for the trajectory tracking with different working conditions. In this case, the desired trajectories were set with two kinds: slow motion trajectory $y_d = \frac{2}{9}\pi[0.5 - \cos(\pi t)][1 - \exp(-t)]$ rad, and fast motion trajectory $y_d = \frac{2}{9}\pi[0.5 - \cos(4\pi t)][1 - \exp(-t)]$ rad. The physical parameters of the joint were set as Equation (41) with the nominal values given in Table 1 (i.e., there exist dynamic uncertainties), and the design parameters of the proposed controller and the ESO were set as the values shown in Table 4. Suppose unknown external disturbances, including the oil stirring loss $T_v$, the external disturbance torque $T_d$, and output shaft dynamic seal friction $T_f$, are also acted on the joint actuator, of which the expressions are expressed in Equations (6), (40) and (42). The position-tracking performance of the slow motion and fast motion are illustrated in Figures 5 and 6 and Table 5, in which the subscripts 'n' and 's' of the figure legends indicate the normal laboratory working condition and deep-sea working condition with 1000 m depth, respectively.

**Table 5.** The maximum steady-state tracking error of the PP-NFTSMC-ESO controllers under different working conditions.

| | Normal Laboratory Working Condition | Deep-Sea Working Condition |
|---|---|---|
| slow motion | $2.55 \times 10^{-4}$ rad | $4.89 \times 10^{-4}$ rad |
| fast motion | $4.36 \times 10^{-4}$ rad | $6.96 \times 10^{-4}$ rad |

As seen, the proposed PP-NFTSMC-ESO controller has excellent position-tracking performance in both slow motion and fast motion with different working conditions. Figures 5a and 6a demonstrate that the actual position response almost overlaps with the desired trajectories. Although the two working conditions impose different influences on the joint actuator and dynamic uncertainties and unknown disturbances vary with time and system states, the position tracking errors were always kept in a very small value (presented in Table 5, with a maximum steady state tracking error about $4.89 \times 10^{-4}$ rad for slow motion and $6.96 \times 10^{-4}$ rad for fast motion) and maintained in the predefined performance boundaries, which are shown in Figures 5b and 6b. As far as position-tracking performance is concerned, the effectiveness of the proposed PP-NFTSMC-ESO controller was clearly shown. The reason for such a good tracking performance may be explained by the fact that, with the help of the ESO, the unmeasured velocity and lumped disturbances (dynamic uncertainties and unknown external disturbances) of the joint actuator can be precisely estimated, which are depicted in Figure 5c,d and Figure 6c,d, and thus the proposed PP-NFTSMC-ESO controller is able to accomplish by employing both the system

model information to achieve the model-based compensation and the additional estimated disturbance signal to reduce the uncompensated effects.

Case 2: This case is to demonstrate the superiority of the proposed PP-NFTSMC-ESO controller for the trajectory tracking. In this case, the proposed controller was compared with three other controllers (i.e., PI, SMC-ESO, and NFTSMC-ESO), which are described in Section 5.2. The physical parameters of the joint actuator, the dynamic uncertainties, and unknown external disturbances were set the same as in Case 1, and the design parameters of the four controllers and the ESO were set as the values shown in Table 4. As the fast motion with deep-sea working condition is more challenging, only the tracking performance of the fast motion trajectory $y_d = \frac{2}{9}\pi[0.5 - \cos(4\pi t)][1 - \exp(-t)]$ rad with the deep-sea working condition was compared, and the simulation resulted with the four controllers presented in Figure 7 and Table 6.

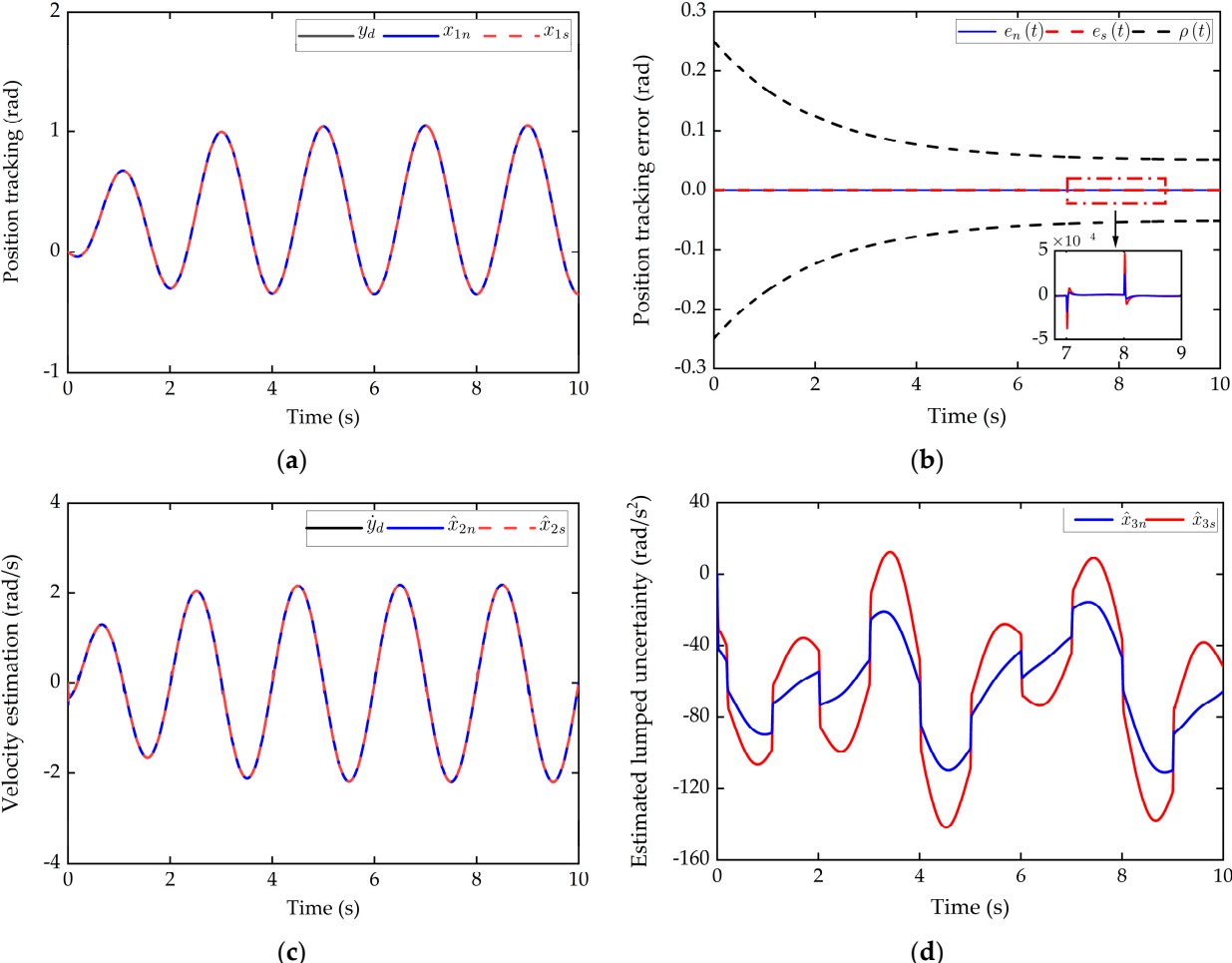

**Figure 5.** Position-tracking performance of slow motion with the proposed PP-NFTSMC-ESO controller. (**a**) Position response. (**b**) Position-tracking error. (**c**) Velocity estimation. (**d**) General disturbance estimation.

**Table 6.** The maximum steady-state tracking error of the four controllers.

| Controller | Maximum Steady State Tracking Error (rad) |
| --- | --- |
| PI | $1.84 \times 10^{-2}$ |
| SMC-ESO | $1.71 \times 10^{-2}$ |
| NFTSMC-ESO | $6.64 \times 10^{-3}$ |
| PP-NFTSMC-ESO | $6.96 \times 10^{-4}$ |

Figure 7a illustrates the position response. On the surface, it seems that almost all the four controllers manifested the same position-tracking performance, but actually they exhibited significant differences from a micro perspective, which can be clearly shown from the position tracking errors in Figure 7b. As seen, the proposed PP-NFTSMC-ESO and NFTSMC-ESO controllers have better tracking performance than PID and SMC-ESO controllers in the overall dynamic response process. This is attributed to the fact that the proposed PP-NFTSMC-ESO controller and NFTSMC-ESO controller not only employ both the system model information to achieve the model-based compensation and the additional estimated disturbance signal to reduce the uncompensated effects, as explained previously, but also take advantage of the property of the fast convergence speed and high stability precision of the NFTSMC controller. In addition, we can also see from Figure 7b that NFTSMC-ESO controller has better tracking performance than the SMC-ESO controller, which is mainly due to the fact that the NFTSMC controller is the improvement of the SMC controller, thus has a faster convergence speed and higher tracking accuracy [30].

Furthermore, it can be seen from Figure 7b that the PP-NFTSMC-ESO controller has superior tracking performance relative to the NFTSMC-ESO controller, of which the maximum steady state tracking error reduces almost 10 times compared with the NFTSMC-ESO controller shown in Table 6. This is mainly due to the fact that introducing the prescribed performance function into the controller makes it more efficient to improve the control action, which can be observed from the control input signal presented in Figure 7c.

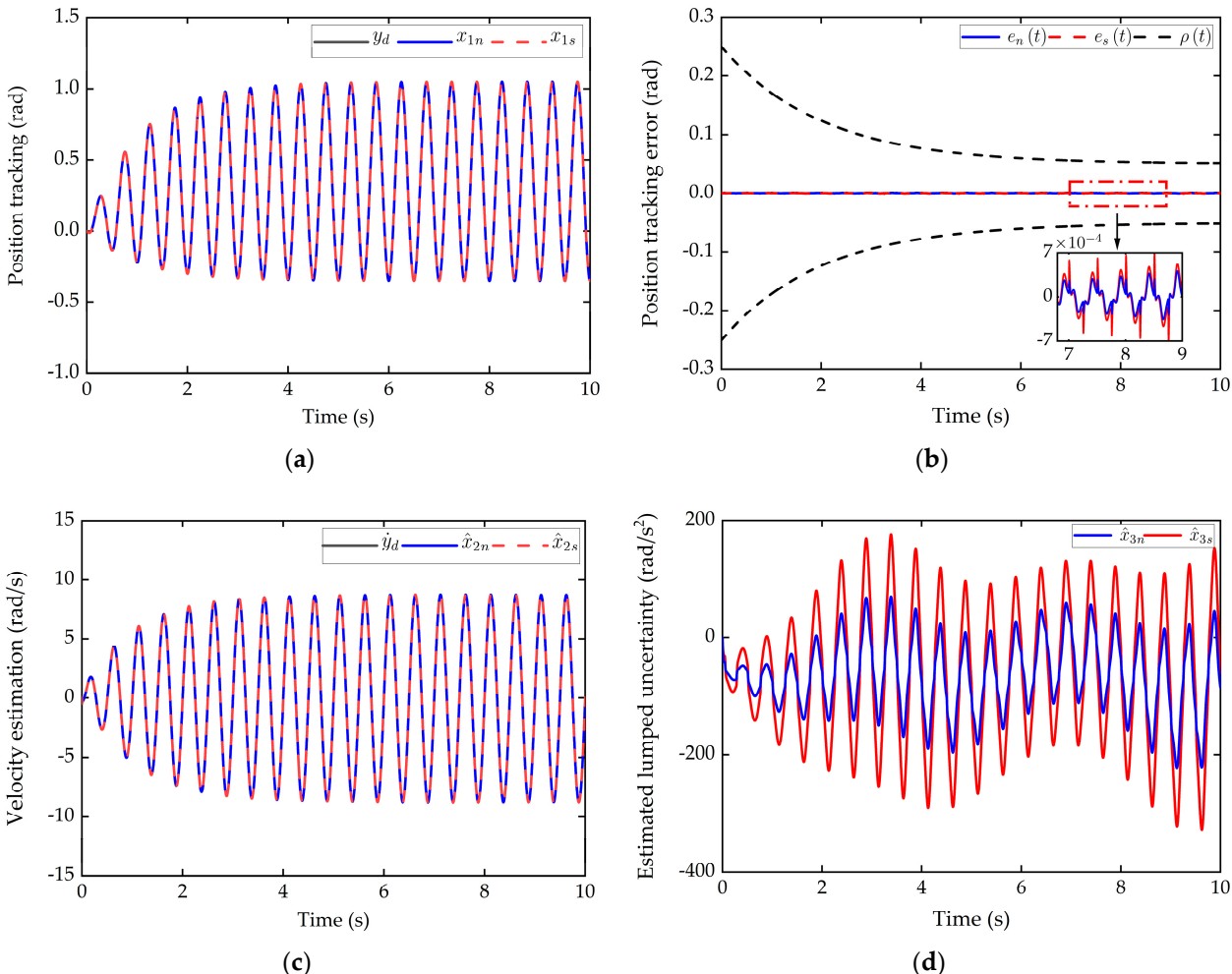

**Figure 6.** Position-tracking performance of fast motion with the proposed PP-NFTSMC-ESO controller. (**a**) Position response. (**b**) Position-tracking error. (**c**) Velocity estimation. (**d**) General disturbance estimation.

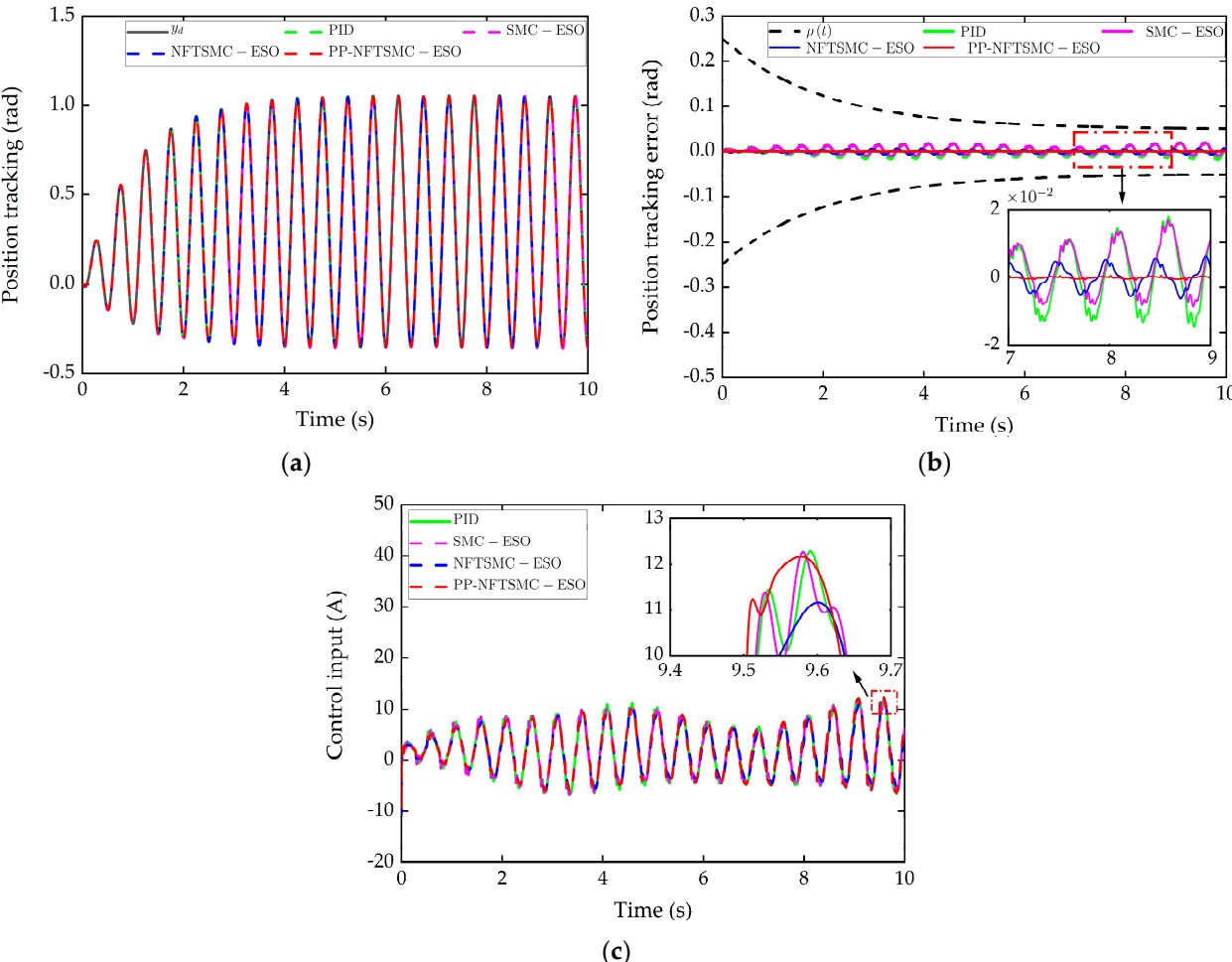

**Figure 7.** Position-tracking performance of fast motion with the four controllers. (**a**) Position response. (**b**) Position-tracking error. (**c**) Control input.

## 6. Conclusions

In this paper, a novel PP-NFTSMC-ESO method was proposed for the high-performance trajectory tracking control of a deep-sea electric oil-filled joint actuator in the presents of unmeasured system states, dynamic uncertainties, and unknown disturbances. The proposed controller integrates the powerful estimation capacity of ESO on immeasurable states and lumped uncertainties and the excellent constraint ability of PPF on the instantaneous and steady-state performance of tracking error with the outstanding performance of the NFTSMC control through backstepping method. The stability of the closed-loop control is guaranteed by the Lyapunov theory. Comparative simulation studies were carried out to demonstrate the effectiveness and superiority of the proposed control scheme.

**Author Contributions:** Conceptualization, L.L. and T.T.; methodology, L.L. and B.L.; software, Y.W. and T.T.; validation, B.L. and G.Y.; formal analysis, L.L.; investigation, L.L. and D.Z.; resources, Y.W. and D.Z.; data curation, D.Z.; writing—original draft preparation, L.L. and Y.W.; writing—review and editing, G.Y., Y.W. and T.T.; visualization, D.Z. and T.T.; supervision, B.L.; project administration, B.L. and G.Y.; funding acquisition, G.Y. All authors have read and agreed to the published version of the manuscript.

**Funding:** This research was funded by the National Key Research and Development Program of China (Grant No. 2016YFC0301700).

**Institutional Review Board Statement:** Not applicable.

**Informed Consent Statement:** Not applicable.

**Data Availability Statement:** Not applicable.

**Acknowledgments:** This work was supported by the researchers (Xinliang Wang, Zhi Liu, Dinfeng Liu, Kang Zhang, etc.) of the Second Ship Design and Research Institute of Wuhan for their technical support.

**Conflicts of Interest:** All authors announce that they have no conflict of interest in relation to the publication of this article.

## Appendix A

The design of the SMC-ESO controller: Define the position error $z_1 = x_1 - x_{1d}$ and $z_2 = \hat{x}_2 - \dot{x}_{1d}$, and selected the SMC surface as

$$\sigma = z_2 + k_1 z_1 \tag{A1}$$

where $k_1$ is a positive design parameter.

The time derivative of $\sigma$ is

$$
\begin{aligned}
\dot{\sigma} &= \dot{\hat{x}}_2 - \ddot{x}_{1d} + k_1 \dot{z}_1 \\
&= \dot{x}_2 - \dot{\tilde{x}}_2 - \ddot{x}_{1d} + k_1 \dot{z}_1 \\
&= \theta_{1n} u + \theta_{2n} x_2 + x_3 - \left(\theta_{2n}\tilde{x}_2 + \tilde{x}_3 - l_2\omega_0^2\tilde{x}_1\right) - \ddot{x}_{1d} + k_1\left(\hat{x}_2 + \tilde{x}_2 - \dot{x}_{1d}\right)
\end{aligned}
\tag{A2}
$$

The control $u$ of SMC-ESO controller is designed as

$$u = -\frac{1}{\theta_{1n}}\left(u_{eq} + u_{sw}\right) \tag{A3}$$

$$u_{eq} = \theta_{2n}\hat{x}_2 + \hat{x}_3 + k_1\left(\hat{x}_2 - \dot{x}_{1d}\right) - \ddot{x}_{1d} \tag{A4}$$

$$u_{sw} = \eta \text{sign}(\sigma) + k_2\sigma \tag{A5}$$

where $k_2$ and $\eta$ are positive design parameters.

Adding Equations (A3)–(A5) into (A2) gives

$$\dot{\sigma} = l_2\omega_0^2\tilde{x}_1 + k_1\tilde{x}_2 - \eta\text{sign}(\sigma) - k_2\sigma \tag{A6}$$

Define the Lyapunov function as

$$V = \frac{1}{2}\sigma^2 \tag{A7}$$

According to Equation (A6), the time derivative of $V$ is

$$
\begin{aligned}
\dot{V} &= \sigma\left[l_2\omega_0^2\tilde{x}_1 + k_1\tilde{x}_2 - \eta\text{sign}(\sigma) - k_2\sigma\right] \\
&= l_2\omega_0^2\tilde{x}_1\sigma + k_1\tilde{x}_2\sigma - \eta|\sigma| - k_2\sigma^2 \\
&\leq \frac{1}{2}\left(l_2\omega_0^2\tilde{x}_1\right)^2 + \frac{1}{2}\sigma^2 + \frac{1}{2}(k_1\tilde{x}_2)^2 + \frac{1}{2}\sigma^2 - \eta|\sigma| - k_2\sigma^2 \\
&= -(k_2 - 1)\sigma^2 - \eta|\sigma| + \Omega_1
\end{aligned}
\tag{A8}
$$

where $\Omega_1 = \frac{1}{2}l_2^2\omega_0^4\tilde{x}_1^2 + \frac{1}{2}k_1^2\tilde{x}_2^2$.

As $\Omega_1$ is a function of estimation errors, which will converge to a very small value when selected a large enough $\omega_0$, and therefore, the closed-loop system is stable with the proposed control law (A3)–(A5).

## Appendix B

The design of the NFTSMC-ESO controller: Define the position error $z_1 = x_1 - x_{1d}$ and $z_2 = \hat{x}_2 - \dot{x}_{1d}$, and selected the NFTSMC surface as

$$\sigma = z_2 + \int_0^t \left(k_1|z_1|^{[\alpha_1]} + k_2|z_2|^{[\alpha_2]}\right)d\tau \tag{A9}$$

where $\alpha_1$, $\alpha_2$ are positive constants subject to $0 < \alpha_1 < 1$ and $\alpha_1 < \alpha_2$, $|z_i|^{[\alpha_i]} = |z_i|^{\alpha_i}\mathrm{sgn}(z_i)$, $i = 1$, 2; $k_1$ and $k_2$ are positive design parameters.

The time derivative of $\sigma$ is

$$
\begin{aligned}
\dot{\sigma} &= \dot{\hat{x}}_2 - \ddot{x}_{1\mathrm{d}} + k_1|z_1|^{[\alpha_1]} + k_2|z_2|^{[\alpha_2]} \\
&= \dot{x}_2 - \dot{\tilde{x}}_2 - \ddot{x}_{1\mathrm{d}} + k_1|z_1|^{[\alpha_1]} + k_2|z_2|^{[\alpha_2]} \\
&= \theta_{1n}u + \theta_{2n}x_2 + x_3 - \left(\theta_{2n}\tilde{x}_2 + \tilde{x}_3 - l_2\omega_0^2\tilde{x}_1\right) - \ddot{x}_{1\mathrm{d}} + k_1|z_1|^{[\alpha_1]} + k_2|z_2|^{[\alpha_2]}
\end{aligned}
\tag{A10}
$$

The control u of NFTSMC-ESO controller is designed as

$$
u = -\frac{1}{\theta_{1n}}\left(u_{eq} + u_{sw}\right)
\tag{A11}
$$

$$
u_{eq} = \theta_{2n}\hat{x}_2 + \hat{x}_3 - \ddot{x}_{1\mathrm{d}} + k_1|z_1|^{[\alpha_1]} + k_2|z_2|^{[\alpha_2]}
\tag{A12}
$$

$$
u_{sw} = \eta\,\mathrm{sign}(\sigma) + k_2\sigma
\tag{A13}
$$

where $k_2$ and $\eta$ are positive design parameters.

Adding Equations (A11)–(A13) into (A10) gives

$$
\dot{\sigma} = l_2\omega_0^2\tilde{x}_1 - \eta\,\mathrm{sign}(\sigma) - k_2\sigma
\tag{A14}
$$

Define the Lyapunov function as

$$
V = \frac{1}{2}\sigma^2
\tag{A15}
$$

From Equation (A14), the time derivative of $V$ is

$$
\begin{aligned}
\dot{V} &= \sigma\left[l_2\omega_0^2\tilde{x}_1 - \eta\,\mathrm{sign}(\sigma) - k_2\sigma\right] \\
&= l_2\omega_0^2\tilde{x}_1\sigma - \eta|\sigma| - k_2\sigma^2 \\
&\leq \frac{1}{2}\left(l_2\omega_0^2\tilde{x}_1\right)^2 + \frac{1}{2}\sigma^2 - \eta|\sigma| - k_2\sigma^2 \\
&= -\left(k_2 - \frac{1}{2}\right)\sigma^2 - \eta|\sigma| + \Omega_2
\end{aligned}
\tag{A16}
$$

where $\Omega_2 = \frac{1}{2}l_2^2\omega_0^4\tilde{x}_1^2$.

As explained previously, $\Omega_2$ is also a function of the estimation error, which will converge to a very small value when selected a large enough $\omega_0$; therefore, the stability of the closed-loop system is obtained with the proposed control law (A11)–(A13).

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
