# Peer review of "Prescribed Performance Non-Singular Fast Terminal Sliding Mode Control Based on Extended State Observer for a Deep-Sea Electric Oil-Filled Joint Actuator"

_applsci, doi:10.3390/app112110130_

Round 1
Reviewer 1 Report
This work proposed a novel observer via robust control method, namely prescribed performance non-singular fast terminal sliding mode control (PP-NFTSMC-ESO) to improve dynamic performance of a deep-sea electric oil-filled joint actuator. Vigorous background is provided with clear aim of the paper. The formulation is sound and a few comparative studies are provided, showing more than 10 times the improvement in steady state tracking error as compared to three other controllers.
This is quite a solid study that clearly showed the superior proposed PP-NFTSMC-ESO control method. It would be interesting to see experimental comparison method, in particular whether the unfavorable factors, as described by the authors, are accurately captured in the simulation setup.
For minor suggestions, Figure 2 image quality can be better. Also the highlighted region in Figure 7 (b) could be enlarged as that is the significant result that conclude to Table 6. It is also difficult to see Figure 7 (c) so enlargement on a particular region will help. Another minor one is P and T on line 303 page 9 should be italic.
In summary, I suggest that the paper to be accepted with minor revision.
Reviewer 2 Report
This manuscript presents a disturbance-observer-based second-order sliding-mode controller for the deep sea electric oil-filled joint actuator. The structure of the manuscript is good and easy to follow. Stability analyses for the controller and the observer have been presented. The parameters selection for controller and observer have also been discussed. Some simulation results have been given. This is a well-written manuscript containing interesting results that merits publication. However, the following issues need to be dealt with:
- The explanations for some acronyms are repeated. For example, PP-NFTSMC-ESO in lines 113, 245.
- In line 172, the sentence “…including modeling uncertainties and unknown internal and external disturbance” is inappropriate since modeling uncertainties can be regarded as parts of unknown internal disturbance.
- In line 221, the authors claims that the ESO can achieve the finite-time convergence of the estimation errors. It is incorrect. The ESO is not a type of finite-time observer and cannot guarantee the convergence of the estimation errors to the origin in a finite time.
- For the simulation tests, the tested PID controller is actually a PI controller. Therefore, please use PI controller rather than PID controller to describe this tested controller.
- In Remark 1, the authors claim that the parameters of all tested controllers are selected by trial and error. Indeed, it is the most widely used method to select the parameters for the sliding-mode controllers in practice. As for the PI controller, however, there are many popular methods with rigorous theoretical demonstrations. Please choose one of them rather than the trail-and-error method to select the parameters of the PI controller and explain why such parameters are selected.
- Regarding References, some important literature about the controller tuning are missing. For example, the bandwidth selection for the ESO is based on (16) in this manuscript, which was given and discussed in the following article. Please add it as reference and highlight that the bandwidth selection is based on the method proposed by this article.
[1] Gao Z. Scaling and bandwidth-parameterization based controller tuning, Proceedings of the 2003 American Control Conference, Denver, USA, 2003, 4989-4996.
Moreover, more papers about disturbance-observer-based second-order sliding mode control and its application published in recent years should be added, such as
[1] L. Zhang, C. Wei, R. Wu, and N. Cui, “Fixed-time extended state observer based non-singular fast terminal sliding mode control for a VTVL reusable launch vehicle,” Aerospace Science and Technology, 2018, 82-83: 70-79.
[2] J. Liu, Y. Gao, X. Su, M. Wack, and L. Wu, “Disturbance-observer-based control for air management of PEM fuel cell systems via sliding mode technique,” IEEE Transactions on Control Systems Technology, 2019, 27(3): 1129-1138.
[3] J. Zhang, H. Wang, Z. Cao, J. Zheng, M. Yu, A. Yazdani, and F. Shahnia, “Fast nonsingular terminal sliding mode control for permanent-magnet linear motor via ELM,” Neural Computing and Applications, 2020, 32: 14447-14457.
[4] Y.-C. Liu, S. Laghrouche, A. N’Diaye, and M. Cirrincione, “Hermite neural network-based second-order sliding-mode control of synchronous reluctance motor drive systems,” Journal of the Franklin Institute, 2021, 358(1): 400-427.
Round 2
Reviewer 2 Report
I am satisfied with the revisions made by the authors. I recommend this manuscript to be published in Applied Sciences.